# Extracting Quantitative Information from Images Taken in the Wild: A Case Study of Two Vicariants of the *Ophrys aveyronensis* Species Complex

**Anais Gibert** [1,*]**, Florian Louty** [1]**, Roselyne Buscail** [2]**, Michel Baguette** [3,4]**, Bertrand Schatz** [5]
**and Joris A. M. Bertrand** [1,*]

[1] Laboratoire Génome et Développement des Plantes (LGDP), UMR5096, Université de Perpignan Via Domitia—CNRS, F-66860 Perpignan, France; florian.louty@etudiant.univ-perp.fr
[2] Centre de Formation et de Recherche sur les Environnements Méditerranéens (CEFREM), UMR5110, Université de Perpignan Via Domitia—CNRS, F-66860 Perpignan, France; buscail@univ-perp.fr
[3] Institut Systématique, Evolution, Biodiversité (ISEB), UMR 7205, Museum National d'Histoire Naturelle, CNRS, Sorbonne Université, EPHE, Universtité des Antilles, F-75005 Paris, France; michel.baguette@mnhn.fr
[4] Station d'Ecologie, Théorique et Experimentale (SETE), UMR 5321, CNRS, Université Toulouse III, F-09200 Moulis, France
[5] CEFE, CNRS, Univiversité Montpellier, EPHE, IRD, F-34293 Montpellier, France; bertrand.schatz@cefe.cnrs.fr
[*] Correspondence: anais.gibert@gmail.com (A.G.); joris.bertrand@univ-perp.fr (J.A.M.B.)

**Abstract:** Characterising phenotypic differentiation is crucial to understand which traits are involved in population divergence and establish the evolutionary scenario underlying the speciation process. Species harbouring a disjunct spatial distribution or cryptic taxa suggest that scientists often fail to detect subtle phenotypic differentiation at first sight. We used image-based analyses coupled with a simple machine learning algorithm to test whether we could distinguish two vicariant population groups of an orchid species complex known to be difficult to tease apart based on morphological criteria. To assess whether these groups can be distinguished on the basis of their phenotypes, and to highlight the traits likely to be the most informative in supporting a putative differentiation, we (i) photographed and measured a set of 109 individuals in the field, (ii) extracted morphometric, colour, and colour pattern information from pictures, and (iii) used random forest algorithms for classification. When combined, field- and image-based information provided identification accuracy of 95%. Interestingly, the variables used by random forests to discriminate the groups were different from those suggested in the literature. Our results demonstrate the interest of field-captured pictures coupled with machine learning classification approaches to improve taxon identification and highlight candidate traits for further eco-evolutionary studies.

**Keywords:** colour quantification; colour pattern analyses; geomorphometry; integrative taxonomy; *Ophrys*; random forest; machine learning

## 1. Introduction

Natural patterns of phenotypic variation are not always congruent with patterns of genetic variation and/or patterns of spatial distribution of species. On the one hand, empirical biogeographic studies have shown that individuals from allopatric populations sometimes appear as phenotypically homogeneous across disjunct geographic ranges, whereas, on the other hand, a careful examination of sympatric individuals sometimes reveals the existence of phenotypically similar but phylogenetically different taxa (i.e., the so-called "cryptic lineages") [1–3]. Although different eco-evolutionary scenarios involving recent vicariance, dispersal, or biological introduction, incipient divergence in sympatry, secondary contact, and/or phenotypic convergence may be invoked to explain such overall phenotypic similarity, a question remains: are we sure that these individuals cannot be distinguished based on their phenotypes? [2,4]. Identifying even a few phenotypic traits

involved in the phenotypic differentiation may at the same time inform taxonomy and give insights into the ecological and evolutionary mechanisms leading to the observed geographic and/or genetic differentiation [3,5]. Recent advances in image-based phenotyping offer new opportunities to capture phenotypic complexity [6] and, therefore, to identify new phenotypic traits to distinguish groups of individuals and better understand the eco-evolutionary causes of phenotypic differentiation.

Over the past decade, image-based approaches have kept their promises in biology, enabling automated species identification in taxonomy [7–9] or high-throughput phenotyping of cultivars in agronomy [10]. In ecology and evolutionary studies, however, image-based phenotyping is still the exception rather than the norm [6]. Systematicists, taxonomists, and evolutionary biologists have rather defined taxa or ecotypes based on qualitative traits (e.g., morph) or a limited set of quantitative traits. However, image-based approaches can provide a quantitative framework for numerous traits, such as colour, patterns, or shapes which have been historically considered as criteria of choice for plant species identification. Consider flower colour as an example. This trait has been often used qualitatively to delineate plant taxa but it may also be quantitatively defined based on a spectrum that summarises the intensity of the different light wavelengths reflected by the flower [11]. Obviously, acquiring such information is cumbersome, especially in the field. As a simple alternative, digital photographs natively capture quantitative colour information as a set of millions of pixels. Each pixel can be registered through 3D coordinates in visual spaces such as RGB or HSV (for Red, Green, and Blue or Hue, Saturation, and Value components, respectively). Until recently, the tools to acquire and analyse such information were limited, so authors tended to use classical image processing software to obtain a proxy of this complex information—for example, by extracting the average colour of a picture by computing its mean values (and associated standard deviations) along the different colour space axes [12]. Fortunately, an increasing number of tools are being made available and provide accessible, automated, and consistent methods for digital image analysis [13–20]. In particular, recent R packages allow us to obtain information and carry on all downstream statistical analyses within the same computing environment [21–25]. Imaging techniques have advanced so far that phenotypic characteristics can now be acquired easily.

Most tools and packages for image analysis allow us both to extract information from images and to visualise variations between individuals/groups (via a heatmap [25] or principal component analysis [21,23]). The output of these packages (raw data, PCA scores, or distance matrices) can also be used to infer statistical differences (e.g., multivariate analyses such as MANOVA, PERMANOVA). A further step might consist in combining information extracted from different packages (colouration, morphometry, etc.) to perform prediction. The democratisation of machine learning approaches allows us to achieve this objective. Among them, the random forest approach is one of the most used classification algorithms [26]; it is simple to implement (in both R and Python languages), fast in operation, able to deal with small sample sizes and high-dimensional feature spaces, and has proven to be extremely accurate for balanced datasets [27]. This method seems particularly promising to test whether or not it is possible to predict, based on phenotypic information, to which group (genetic or geographic group) an individual belongs to.

Here, we used image-based analyses coupled with a simple machine learning classification algorithm to test whether such an approach may allow us to distinguish closely related taxa belonging to the orchid genus *Ophrys*. In *Ophrys*, flower colouration and colour patterning as well as shape are certainly of primary importance in the diversification process. The *Ophrys* genus diversified in the Euro-Mediterranean region, where it experienced an explosive evolutionary radiation with the formation of tens to hundreds of lineages over the last five million years, thus harbouring one of the top diversification rates reported so far [28]. This extraordinary diversity was shown to be related to the relationship that these plants maintain with their pollinator insects, and their underlying highly specialised pollination syndrome: sexual swindling (see [29] for a recent review). Indeed, the *Ophrys* flower emits a blend of volatile compounds that is similar to the sexual pheromones pro-

duced by receptive female insects (and is therefore often referred to as pseudo-pheromones). The male is attracted by this olfactive stimulus, but visual and morphological stimuli are also involved in the sequence. The aspect of the flower mimics the colour and the pilosity of the female insect and will provoke the landing of the attracted male, which will eventually try to copulate (the so-called pseudo-copulation) with the flower. These visual and morphological stimuli are crucial for the success of *Ophrys* pollination as they are primarily involved in the appropriate positioning of the insect male on the flower to cause pollen to be glued to its body (on the insect head or abdomen depending on the *Ophrys* species considered). A close mechanical fit between *Ophrys* flowers and their male pollinators during pseudo-copulation is expected on ensuring the effective pollinia transfer [30]. This will provide the opportunity to the insect to transfer pollen to another individual plant (which it sees as a new potential sexual partner). For many *Ophrys* species, the similarity between the plant and the shape, the dimension, the hairiness, and the colouration of the insect is obvious [30,31]. Several studies have demonstrated it experimentally [32], whereas others have shown that the colour values and colouration patterns of the flower can be perceived and memorised by the pollinator, thus having ecological and evolutionary significance in this particular system ([33–37], but see [38]). The evolution of these sexually deceptive orchids [28] favours intraspecific variation in floral morphology and colours, which often makes it difficult to delimit each species and to assign the most divergent individuals to a species. This is why the development of decision support tools for this assignment is particularly relevant for the fundamental knowledge and conservation of certain protected species among these *Ophrys*.

The *Ophrys aveyronensis* system species complex seems relevant to test our approach. This species complex is a typical example of a taxon with disjunct geographic distribution [39]. It consists of two vicariants separated by 600 km that are (i) phenotypically similar (in Appendix A Figure A1) and (ii) phenotypically variable at the inter-individual level (in Appendix A Table A2). This situation seems particularly challenging for image-based taxa identification [8]. In addition, all apps for plant identification are today unable to provide subspecies identification for this taxon. With this example as a case study, we aimed at testing the feasibility and the efficiency of a research approach consisting of (i) deriving quantitative information from pictures taken in field conditions, (ii) using easy-to-use existing tools in the R statistical computing environment to process this information, and (iii) feeding a machine learning tool (i.e., random forest models) with this information in order to see whether and to what extent we could discriminate the two vicariants of the *Ophrys aveyronensis* species complex. We then discuss the interest of this approach for integrative taxonomy [12], for the interpretation of particular biogeographic patterns, and to identify eco-evolutionarily relevant candidate traits.

## 2. Materials and Methods

### 2.1. Study System

*Ophrys sphegodes* Mill. subsp. *aveyronensis* [40] or *Ophrys aveyronensis* [41] is an *Ophrys* species that was initially described as endemic to a geographically restricted area in the 'Grands Causses' region (Southern France). Ever since, phenotypically similar populations have been discovered sporadically, in the North of Spain [42–44]. These Iberian populations have been considered as a distinct taxon: *Ophrys vitorica* by some authors [39,45]. However, considering that the same insect species (a solitary bee: *Andrena hattorfiana*) was shown to be a pollinator shared by all the populations and pointing out that the morphometry and the ecology of the plant appear similar across its range, Paulus rather proposed to consider the Iberian populations as *Ophrys aveyronensis* subsp. *vitorica* [46]. Here, we refer to the two taxa of this complex as subspecies: *O. aveyronensis* subsp. *aveyronensis* in France, and *O. aveyronensis* subsp. *vitorica* in Spain.

## 2.2. Collection Sites and Plant Material

In June 2019, we collected samples from a total of 109 individuals at 6 localities representative of the disjunct geographic distribution of the *Ophrys aveyronensis* species complex: 3 sites in France (Guilhaumard, Lapanouse-de-Cernon, and Saint-Affrique) for *O. aveyronensis* subsp. *aveyronensis* and 3 in Spain (Valgañón, Larraona, and Bercedo) for *O. aveyronensis* subsp. *vitorica* (Figure 1). From each sampling site, between 13 and 22 individuals were identified, measured, and photographed (Table A1). A total of 11 phenotypic traits (and pictures) were recorded for 52 individuals of *O. a.* subsp. *aveyronensis* and 57 individuals of *O. a.* subsp. *vitorica*). As *Ophrys aveyronensis* is legally protected in France, we asked for and obtained a "Demande de dérogation pour la récolte, l'utilisation et le transport de spécimens d'espèces végétales protégées" (arrêté préfectoral 2019-s-16, on 7 May 2019) from the Direction Régionale de l'Environnement , de l'Aménagement et du Logement(DREAL) Occitanie. *Ophrys aveyronensis* is not protected in Spain.

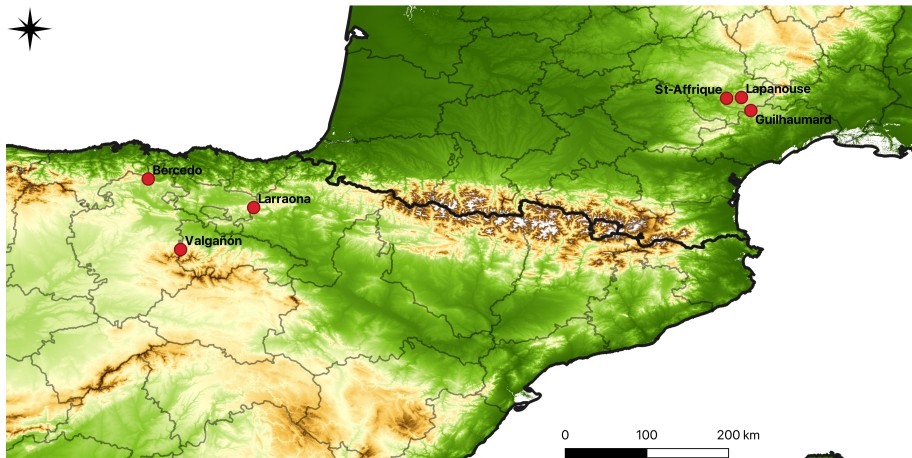

**Figure 1.** Map of *Ophrys aveyronensis* populations. *O. aveyronensis* populations were sampled across the geographic range of the species in Southern France and Northern Spain.

## 2.3. Morphological Traits Measured in the Field

In each field site, we recorded five "whole-plant" traits on each individual: plant size, stem diameter, number of reproductive structures (number of flowers + buds), distance from the ground to the first flower, and distance between the first and the second flower. These traits reflect the plant's biomass and fecundity. We also measured six floral traits: length and width of the labellum, right petal, and median sepal. Length was measured from the insertion area to the top of the organ. Width was systematically measured at the largest part of the organ. All these traits (except the count of reproductive structures) were measured with a ruler or a calliper. Floral traits (morphology and colour at least) are of particular importance for biotic interaction, especially for pollinator attraction, and for the close mechanical fit between *Ophrys* flowers and their male pollinators; a difference in these floral traits could therefore reflect a potential difference in pollinators between these subspecies. This dataset is called the "Morphology (field-based)" dataset in the following analysis (see distribution of traits in Appendix A Figure A2).

## 2.4. Information Extracted from Images
### 2.4.1. Picture Acquisition

The (raw) photographic database was generated during the fieldwork session. All photographed specimens were first identified as *O. aveyronensis* (subsp. *aveyronensis* in France and subsp. *vitorica* in Spain) based on morphological taxonomic criteria. All the pictures were taken following three criteria for calibration: (i) same material (camera and lens) and settings (ISO, aperture value, and shutter speed), (ii) same magnification ratio, and (iii) the use of a ring flash to obtain a homogeneous colour temperature. We used a

Canon EOS 70D camera, an EF 100 mm F/2.8 USM macro lens, and a Metz Mecablitz MS-15 digital ring flash with a magnification ratio of 2:3, an aperture value of F/18, a shutter speed of 1/250 s, and an ISO sensitivity of 100. On each picture, a Truecolors Scuadra grey chart allowed us to adjust the white balance during the post-processing.

### 2.4.2. Post-Processing and Filtering of the Photographic Database Elements

On each picture, the white balance was first adjusted by setting the colour temperature to 5600 Kelvin (corresponding to the colour temperature emitted by the ring flash we used) before undergoing file conversion from raw (Canon .cr2) to .jpeg (Adobe RGB 1998) in Adobe Lightroom. The background was removed for each picture thanks to the 'smart scissors' tool in the software Gimp 2.10.4 (http://www.gimp.org, accessed on 1 May 2020). In addition to this whole-flower (without background) dataset, we also generated a picture dataset comprising labellum only.

### 2.4.3. Morphological Traits Extracted from Images

We manually measured six traits on each whole-flower image specimen (the distance between pseudo-eyes, the length and width of the labellum, median sepal, and right petal) by using the "ruler" tool of GIMP 2.10.4. This dataset is called the "Morphology (image)" dataset in the following analysis.

### 2.4.4. Flower Shape

We extracted information about flower shape using a common approach to perform shape analysis called geometric morphometrics. This approach uses the two-dimensional coordinates of landmarks to record the relative positions of morphological points as the basis of shape variation quantification. We followed the steps described in Adams and Otarola-Castillo [21]). First, we defined 13 two-dimensional landmark coordinates on *Ophrys* flowers (Figure 2); these landmarks record the relative position of anatomically corresponding locations such as the tip of each sepal and petal, the position of each pseudo-eye, the labellum inlets, etc. We used TpsDig2 software to manually set up the 13 landmarks on each whole-flower image specimen [47]. Next, we used the R package *geomorph* [21] to perform a generalised Procrustes analysis. This analysis superimposes the flowers to a common coordinate system by holding constant variation in their position, size, and orientation. Finally, we performed a principal component analysis (PCA) on the Procrustes-aligned coordinates (x,y) of the 13 landmarks and reported the coordinates of each specimen from the five first axes of PCA. These coordinates were used in the following analysis under the name "Flower shape" dataset.

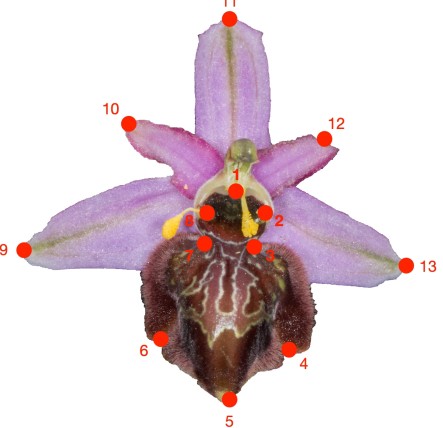

**Figure 2.** Positions of the 13 landmarks on an *Ophrys aveyronensis* flower. 1: Top of stigmatic cavity; 2-8: Position of pseudo-eyes; 3-4-6-7: Labellum inlets; 5: Tip of the labellum; 9-11-13: Tips of each sepal; 10-12: Tips of each petal.

### 2.4.5. Flower Colour

We extracted flower colour information on each whole-flower image specimen using the recent R package *colordistance* [25]. This package provides an objective and easy-to-use alternative to manual classifications of colour. For simplicity, we used the histogram binning method in the Red–Green–Blue colour space (RGB), but other methods (e.g., *k*-means) and colour spaces are also available. In our case, the histogram binning method was preferred over the *k*-means binning one because it ensures consistent comparisons among images. First, the package treats each pixel as 3D coordinates in RGB colour space (Figure 3a,b). Next, we set up a number of colour bins, based on our estimation of different colours in our image. Here, we used 8 colour bins (the minimum: each of the red, green, and blue axes being divided into 2 bins), but we also ran the procedure with 27 (each RGB axis divided by 3), 64 (divided by 4), and 125 (divided by 5) bins (Figure 3) and the results were similar. Next, the algorithm counts how many pixels fall into each of these predetermined bins for each image specimen. Comparisons were thus consistent across images because the bins had the exact same bounds among images. Finally, the package provided a colour histogram for each image (Figure 3c,f). This dataset, composed of the proportion of pixels in each of the 8 bins of colour for each individual, was called the "Whole-flower colour" dataset (see distribution of pixels by bin of colour in Appendix A Figure A3).

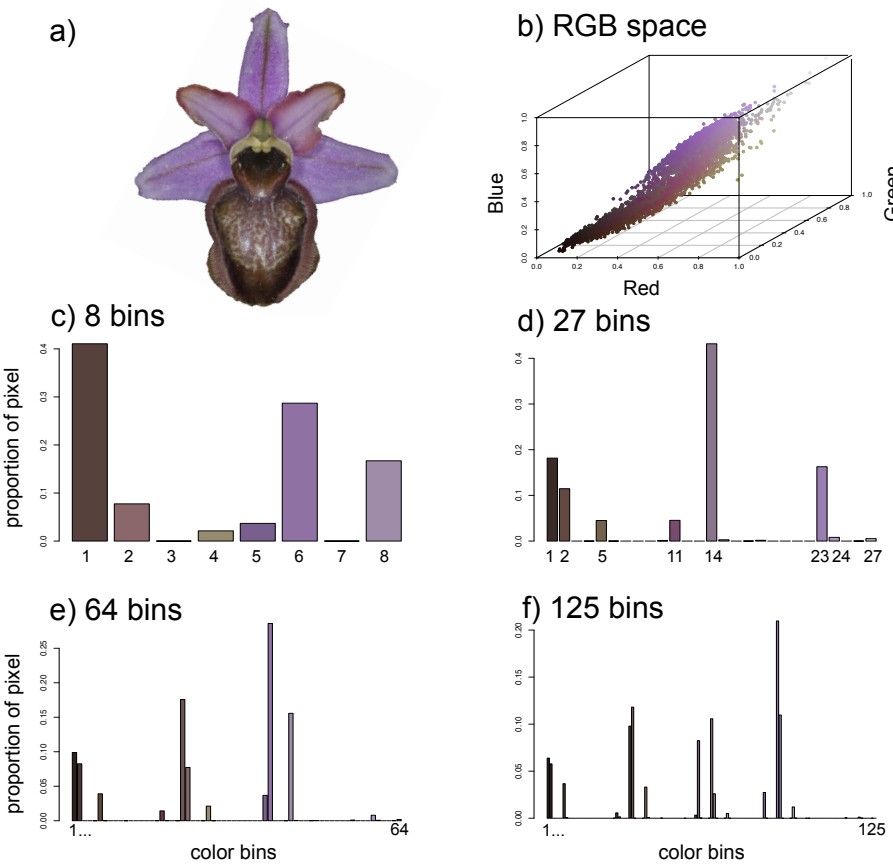

**Figure 3.** Colour binning for an image specimen of *Ophrys aveyronensis* subsp. *aveyronensis*. (**a**) Whole-flower image. (**b**) Scatterplot of all non-background pixels in RGB space. (**c**) Histogram showing the proportion of pixels assigned to each of eight bins, (**d**) 27 bins, (**e**) 64 bins, and (**f**) 125 bins. For plots c–f, bins are coloured by the average colours of the pixels in each bin, and X and Y axes refer to the colour bins and the proportion of pixels by image, respectively.

### 2.4.6. Labellum Colour Pattern

*Ophrys aveyronensis* shows natural variation in the labellum colouration pattern (Figure 4a); some individuals exhibit vivid and clearly delineated line patterns contrasting with the dark labellum ("H-like" pattern), whereas others harbour more marbled patterns with greenish colours ("marbled" pattern). Measuring variation in colour patterns consisted in analysing the spatial distribution of colours in the image. Therefore, consistent comparisons across image specimens require (i) an alignment of images on a homologous anatomical structure and (ii) a colour-based segmentation of images. Here, we used the R package *patternize* [23], which combines both image transformation and colour extraction approaches. First, we used the automated image registration technique to transform the image. We chose this method for its relative simplicity, but other alignment methods (such as landmark-based ones) are also available in this package. Next, we used two different approaches for colour extraction: *k*-means clustering and watershed approaches.

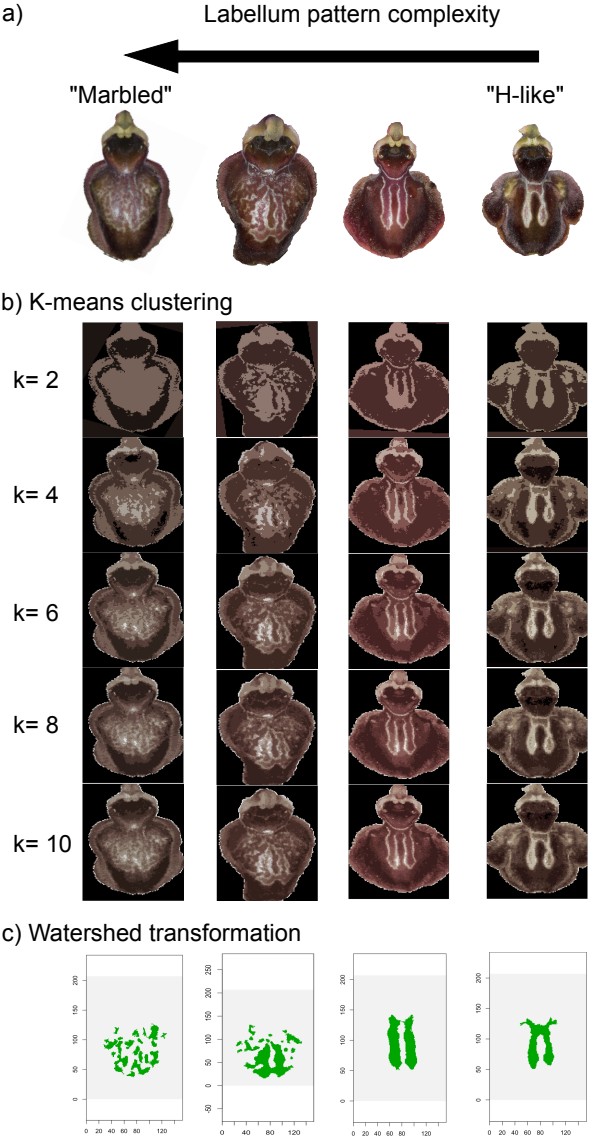

**Figure 4.** Comparisons of image colour extraction with *patternize* using *k*-means clustering and watershed transformation for four specimens of *Ophrys aveyronensis*. (**a**) Labellum image. Specimens exhibited a gradient of complexity from "H-like" to a "marbled" pattern on their labellum. (**b**) *k*-means clustering with *k* = 2, 4, 6, 8, and 10. (**c**) Watershed transformation.

*K*-means clustering is an unsupervised approach for colour-based image segmentation assigning each pixel to k clusters based on RGB values. The number of clusters (k) has to be assigned manually (here, *k* = 6, which was a good compromise between a good assignment of pixels to colour patterns and computational time; see Figure 4a,b). The RGB value of each cluster is first obtained from a reference image (randomly chosen), and then used as a cluster centre for the *k*-means clustering of the subsequently analysed images. In the reference image, each pixel in the image is assigned to the RGB cluster that minimises the distance between the pixel and the cluster centres, and at each iteration, the cluster centre is recalculated by averaging all pixels in the cluster until convergence. Finally, we performed PCA on pixel matrices (quoted as 0 or 1 depending on whether or not it fell into the cluster under consideration) for each colour cluster, and we reported the coordinates of each specimen from the three first axes of PCA by cluster. These coordinates were used in the following analysis under the name "Labellum colour pattern (K)" dataset.

In watershed transformation, the image is treated as a topographic map based on pixel value changes (gradient map). First, at least one pattern and one background pixel are manually identified on each image specimen. Next, a flooding process propagates pattern and background labels guided by the gradient map until the pattern and background labels meet, determining the watershed lines that are used to segment the image. We then performed a PCA on the pixel matrix and reported the coordinates of each specimen from the six first axes of PCA. These coordinates were used in the following analysis under the name "Labellum colour pattern (W)" dataset.

The advantage of the watershed approach is that manual assignments allow us to confidently identify the pattern on the labellum (continuous gradient between "marbled" or "H-like" pattern; see Figure 4c) in the analysed images. The advantage of *k*-means clustering is that it may allow us to highlight the importance of pattern variation that can be considered as unexpected, or on which the observer may have been subjectively not focused, such as marginal pilosity (see Figure 5).

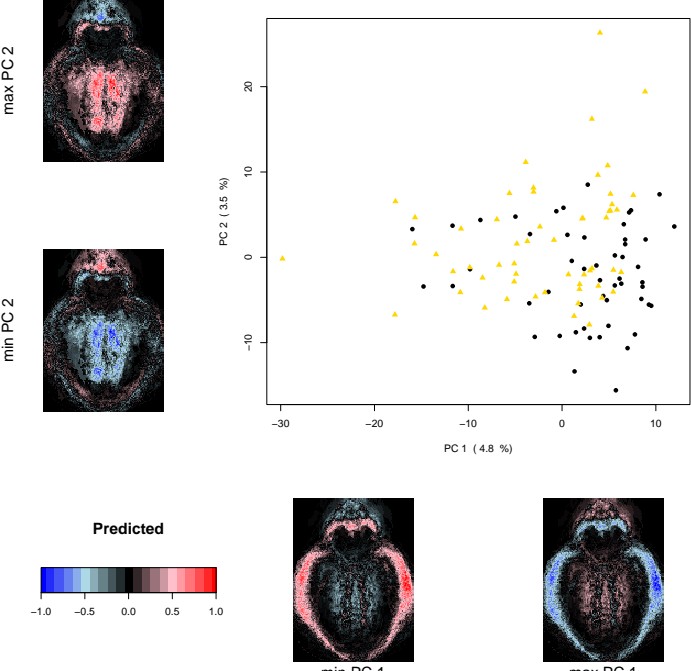

**Figure 5.** Two first axes of principal component analysis (PCA) performed on pixel matrix for cluster 6 (light brown colour, RGB: 141, 132, 109) suggested that *O. a.* subsp. *aveyronensis* and *O. aveyronensis* subsp. *vitorica* differentiate phenotypically based on the presence or absence of labellum marginal pilosity (gold: *O. aveyronensis* subsp. *vitorica*, black: *O. a.* subsp. *aveyronensis*). Pixel matrix for cluster 6 was provided by R package *patternize* using *k*-means clustering transformation (*k* = 6).

*2.5. Statistical Analysis*

All statistical analyses were performed using the R.4.1.2 software [48]. As mentioned above, shape, colour, and colour pattern variation analyses were performed using, respectively, *geomorph* [21], *colordistance* [25], and *patternize* [23] R packages.

### 2.5.1. Random Forest (RF)

Random forest (RF) operates, as the name implies, by growing an ensemble of decision trees and letting them vote for the most popular class. A key principle of RF is that it creates an uncorrelated forest of trees whose prediction is more accurate than that of any individual tree. Indeed, each individual tree is built on a different set of data (random sample with replacement from a training set, i.e., bagging) and also uses only a proportion of variables to make decisions. Here, we build seven RFs (also called classifiers), one for each phenotypic dataset (field-based morphology, image-based morphology, flower shape, whole-flower colour, labellum colour pattern (K), labellum colour pattern (W)) and one for a combined dataset. The dependent variable was the subspecies: *O. aveyronensis* subsp. *aveyronensis* and *O. aveyronensis* subsp. *vitorica*. All RFs were implemented in R via the *randomForest* package [49].

For each RF, we performed the following steps. First, we pre-selected variables to avoid multicollinearity (i.e., strong interdependence of explanatory variables); we eliminated variables (plant size and stem diameter in morphology and combined dataset) with correlation values above 0.8 or below −0.8 (Pearson's coefficient). Plant size was correlated with the distance from the ground to the first flower ($\rho$ = 0.92), and stem diameter with the number of reproductive structures (number of flowers + bud, $\rho$ = 0.80). Second, we split the dataset into a training (80% of the data) and a test set (the remaining 20%). This split was similar for all phenotypic datasets so that the RFs were trained on the same individuals for all datasets. Third, we set up three parameters: the number of trees, the number of variables randomly selected at each node (mtry), and the node size. We ran models with 1000 trees, and we selected an optimal value for mtry based on cross-validation performance (the out-of-bag-error estimate, using the tuneRF function in R), and the node size was set to default. Next, we evaluated the accuracy of each classifier as measured of performance, using the separate test set (20% of the data). We also reported the out-of-bag-error (OOB error), the sensitivity (i.e., the percentage of *O. a.* subsp. *aveyronensis* predicted correctly, also called true positive rate—TPR), the specificity (i.e., the percentage of *O. a.* subsp. *vitorica* predicted correctly, also called true negative rate—TNR), and the area under the curve (AUC) for each RF. Finally, we also checked that our results were not dependent on the initial random split of the dataset. To do so, we re-run 100 RFs by dataset using different training and test datasets at each iteration, and we obtained an averaged accuracy with confident interval (95%).

Using the best classifier, we also identified which variables had the most predictive power. We computed two measures of importance for each variable in the random forest for the complete dataset. The first measure was the mean decrease accuracy. It expressed how much accuracy the model loses by excluding each variable. The second measure was the mean decrease in Gini coefficient. It expressed how each variable contributes to the homogeneity of the nodes in the random forest. If the variable is useful, it tends to split nodes into pure single-class nodes. The variables with the highest importance scores are the ones that give the best prediction and contribute the most to the model.

### 2.5.2. Multivariate Analysis

We tested for overall dissimilarity in morphology, shape (geomorphometry), colour, and colour pattern between subspecies and populations (within species) using permutational multivariate analysis of variance (PERMANOVAs, [50]). From each dataset, we derived matrices based on Euclidean distances for the morphological, shape, and pattern datasets [51], or Chi-squared distance for colour data, as advised by the author of the *colordistance* package [25]. Statistical significance was assessed via 999 permutations. These

analyses were implemented using the *vegan* package in R [52]. For the shape dataset, we also performed Procrustes ANOVA with permutation procedures, as advised by the authors of the R package *geomorph*, to assess differences in shape between species and population, using the "procD.lm" function [21].

## 3. Results

Random forest models showed that *O. a.* subsp. *aveyronensis* and *O. a.* subsp. *vitorica* are distinguishable based on phenotype (see in Appendix A Figures A2 and A3 for differences in trait distribution between subspecies). The highest accuracy (95%) in classifying between *O. a.* subsp. *aveyronensis* and *O. a.* subsp. *vitorica* was reached when the combination of all datasets was considered as the training set ("combined dataset", Table 1). The classifiers built exclusively on flower shape or labellum colour pattern (W) showed relatively poor performance (below 65%, Table 1). These results were confirmed when several RF models (100 runs of 1000 trees) were trained and tested on different subsets of individuals (Figure 6); the highest accuracy was reached by the combined dataset, and the flower shape and the labellum colour pattern (W) classifiers showed poor performance. When re-run 100 using different individuals as the training dataset at each iteration, the classifiers still showed high performance (see mean and CI in performance in Figure 6), suggesting the independence of our results from the initial random split of the training/test dataset.

**Table 1.** Performance indices of random forest models for each dataset and for a combined dataset. OOB error = out-of-bag error, sensitivity = true positive rate (TPR) with *O. a. aveyronensis* as the positive class, specificity = true negative rate (TNP) with *O. a. vitorica* as the negative class, and AUC = area under the curve. All random forests were trained (using 80% of the data) and tested (using 20% of the data) on the same individuals.

| Dataset | mtry | Accuracy | Sensitivity (TPR) | Specificity (TNR) | OOBerror | AUC |
|---|---|---|---|---|---|---|
| Morphology (field) | 3 | 0.73 | 0.64 | **0.82** | 33 | 0.78 |
| Morphology (image) | 2 | 0.59 | 0.5 | 0.7 | 33 | 0.72 |
| Flower shape | 2 | 0.59 | 0.5 | 0.64 | 51 | 0.51 |
| Whole-flower colour | 3 | **0.82** | 0.78 | **0.85** | 25 | 0.80 |
| Labellum colour pattern (K) | 4 | 0.68 | 0.62 | 0.71 | 24 | 0.80 |
| Labellum colour pattern (W) | 2 | 0.64 | 0.6 | 0.65 | 39 | 0.66 |
| Combined | 9 | **0.95** | **0.9** | **1** | 15 | 0.94 |

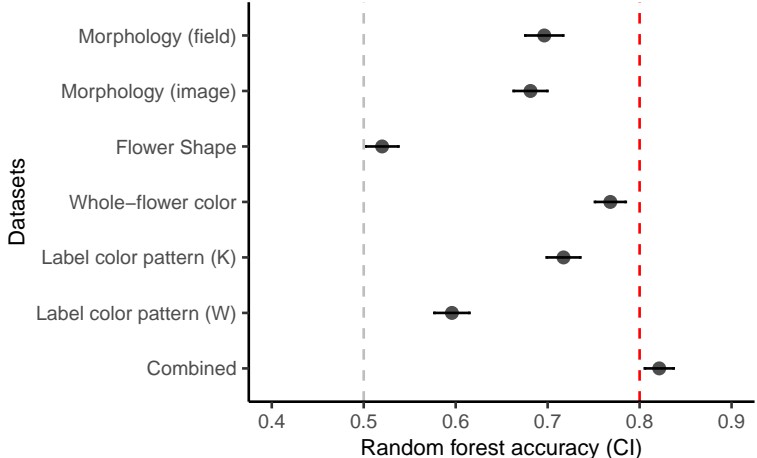

**Figure 6.** Accuracy (mean value, 95% CI) over 100 random forest models for each dataset.

The best classifier (the "combined" one) achieved 100% specificity (i.e., the percentage of *O. a.* subsp. *vitorica* predicted correctly) and had 90% sensitivity (i.e., the percentage of *O. a.* subsp. *aveyronensis* predicted correctly), which outperformed other classifiers. Most classifiers in our study showed higher specificity than sensitivity. This result was

not caused by an unbalanced design, and suggested that *O. a.* subsp. *aveyronensis* was somehow more often misspecified.

In the best classifier, the variable ranking of the 43 variables revealed that—according to both mean decrease accuracy and mean decrease Gini—two colours of whole-flower (bin 5, 3; see Figure 3c), one axis of PCA for labellum colour pattern (K), sepal width, and pseudo-eye distance (Figure 7) were among the five most important variables to distinguish between *O. aveyronensis* subspecies. The axis of PCA for labellum colour pattern (K) explained only a small part of the variance between individuals (PC1: 4.7%) but seemed to distinguish individuals based on their labellum marginal hairiness (Figure 5). Interestingly, labellum width was positioned after sepal width and pseudo-eye distance in the ranking.

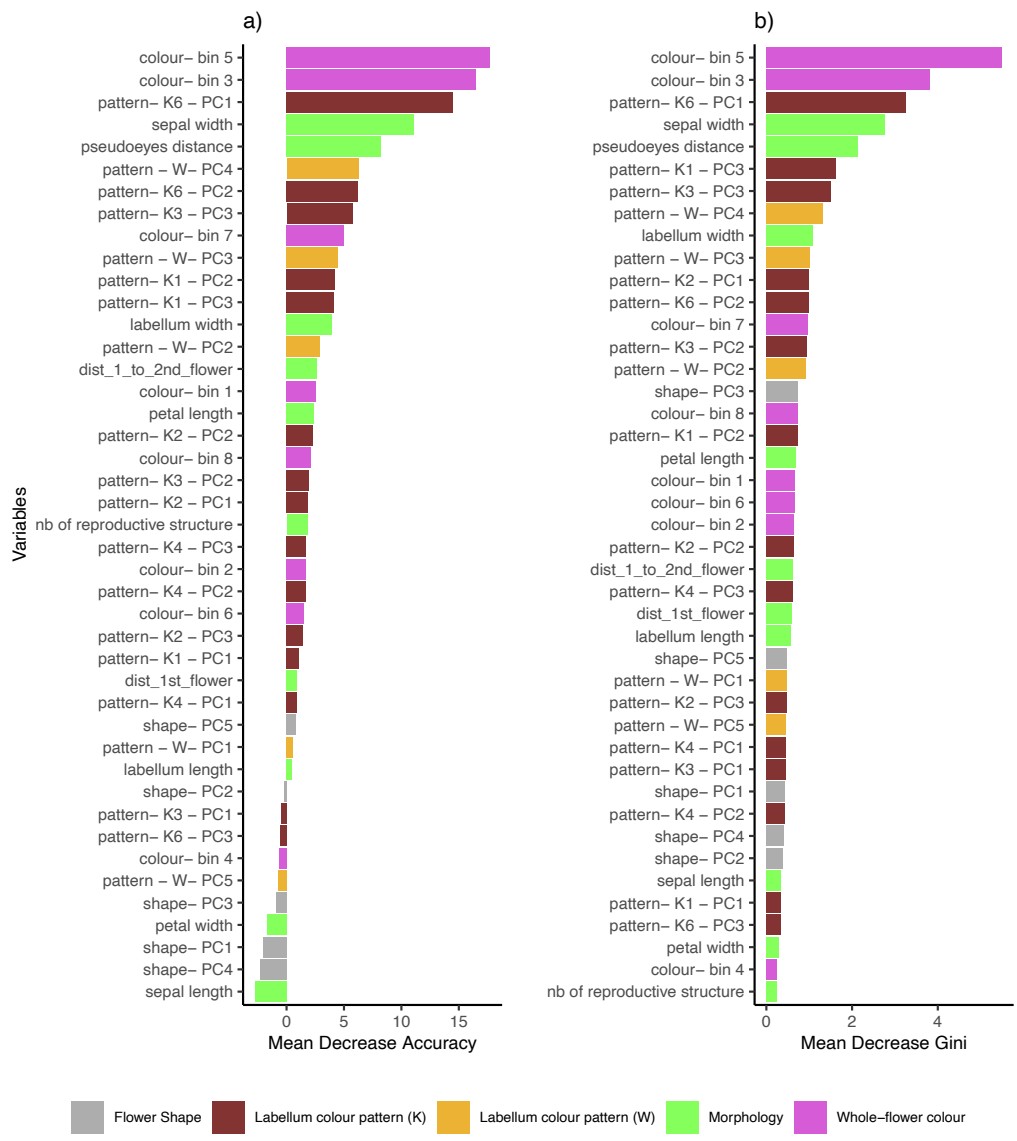

**Figure 7.** Variable importance defined by (**a**) mean decrease accuracy and (**b**) mean decrease gini for distinguishing between *O. a.* subsp. *aveyronensis* and *O. aveyronensis* subsp. *vitorica* based on a random forest with 1000 trees ("Combined" classifier).

Overall, the PERMANOVAs showed differences between subspecies, which is coherent with the RF results (Table 2). The subspecies differed in morphology (image), colour, and colour pattern of labellum (W and K) but not in flower shape or morphology measured in the field. There was also a population effect in most datasets, except in flower shape and morphology (image).

**Table 2.** Phenotypic differences in morphology, flower shape, colour, and labellum colour pattern between *Ophrys aveyronensis* subspecies and populations (within subspecies) as estimated by PERMANOVA analysis.

| Phenotype | Effect | Df | F | *p* |
|---|---|---|---|---|
| Morphology (field) | subspecies | 1 | 1.236 | 0.277 |
| | populations | 4 | 8.861 | **0.001** |
| Morphology (image) | subspecies | 1 | 12.655 | **0.001** |
| | populations | 4 | 1.135 | 0.327 |
| Flower shape | subspecies | 1 | 1.413 | 0.206 |
| | populations | 4 | 1.117 | 0.3 |
| Whole-flower colour | subspecies | 1 | 22.992 | **0.001** |
| | populations | 4 | 11.425 | **0.001** |
| Labellum colour pattern (K) | subspecies | 1 | 1.628 | **0.001** |
| | populations | 4 | 1.266 | **0.001** |
| Labellum colour pattern (W) | subspecies | 1 | 2.222 | **0.001** |
| | populations | 4 | 1.267 | **0.016** |

## 4. Discussion

Our study demonstrates that image-based approaches coupled with random forest algorithms allow us to clearly differentiate the two vicariants of the *O. aveyronensis* species complex (*O. a.* subsp. *aveyronensis* and *O. aveyronensis* subsp. *vitorica*) based on phenotype, and to suggest new candidate traits to investigate the eco-evolutionary causes shaping the subtle phenotypic divergence between these two taxa. Our study thus provides a clear example of the interest of the image-based approach in natural systems, and calls for a more systematic use of these approaches in ecology and evolutionary studies.

While the two subspecies are hardly distinguishable by the human expert's eye, the accuracy of our best classifier reached 95%. This result demonstrates that, when combined, image-based and field-based variables allowed us to perform a reliable classification, and probably to outperform the human capability to assign plant pictures to the correct predefined group (here, subspecies). Here, the most important variables for classification were not specifically derived from a specific type of data, as they were either linked to flower colour, morphological traits, or labellum colour pattern (but not geomorphometry). Interestingly, however, the most relevant colour variables to tease apart the two subspecies were not the colour bins containing the highest proportion of pixels (Figure 3, and in Appendix A Figure A3), suggesting that more subtle differences in flower colour have a relatively high importance in the classification process. Contrary to expectation [39], the labellum pattern criteria—"marbled" and variable labellum pattern for *O. a.* subsp. *aveyronensis* vs. "H-like" and simpler pattern for *O. aveyronensis* subsp. *vitorica*—were poorly informative. This labellum pattern that we specifically highlighted in *patternize* with the watershed segmentation method was even less informative than the variables derived by the semi-automated segmentation method (*k*-mean). We also found two morphological traits to be important for classification: the distance between pseudo-eyes and median sepal width. *O. a.* subsp. *aveyronensis* had on average a larger distance between pseudo-eyes and larger median sepal width than *O. aveyronensis* subsp. *vitorica* (in Appendix A Figure A2). Although there is no mention of the distance between the pseudo-eyes in the literature, this is overall in accordance with Delforge [39], who reported that flower dimensions are overall larger in *O. a.* subsp. *aveyronensis* compared to *O. a.* subsp. *vitorica*. In this way, it should be noted that traits such as labellum length and width, which are usually considered as important in mimicking the pollinator size, were not found to be the most informative for classification in our case. Moreover, due to an overall important overlap in their distribution, none of the traits was found to be diagnostic for taxa delineation. This confirms that it is a combination of traits and a simultaneous consideration of them that allows the success of the classification process.

The plant–insect interaction is highly specific in *Ophrys* but not necessarily one-to-one [53]. Joffard et al. [54] reported an average of 1.56 species of insect pollinator per species

of *Ophrys*. Although one particular species of insect, *Andrena hattorfiana*, was shown to be a common pollinator of both *O. a.* subsp. *aveyronensis* and *O. a.* subsp. *victorica*, the subtle differences that we found in morphometry and colouration suggest that at least one of the two vicariants may attract another insect pollinator species, or that the common pollinator is geographically variable and orchids are locally adapted. This hypothesis of different (or same but variable) pollinators is supported by two types of considerations. Firstly, the two vicariants significantly differ in colouration (in Appendix A Figure A3), suggesting that colour may have been involved in response to distinct insect pollinator preferences. Some traits that are assumed to be important for the insect pollinator attraction and during the pseudo-copulation also display differences in mean trait values and/or distribution shapes (in Appendix A Figure A2). Based on these considerations, and although it will require more investigations, we may speculate that floral traits such as labellum width, petal length, sepal width, and distances between pseudo-eyes may experience different selection pressures and could be involved in an early stage of ecological speciation (in addition to pure allopatry). Secondly, the examination of plant communities that are sympatrically and simultaneously flowering with *O. aveyronensis* show that we have rarely or never observed *Scabiosa columbaria* and *Knautia arvensis* (both from the family *Dipsacaceae* and very close morphologically), which are the typical host plants of the known pollinator of *O. aveyronensis*: *Andrena hattorfiana* [55]. Moreover, these two plants flower between June and August, whereas *O. aveyronensis* flowers between mid-May and early June. This poor flowering synchronisation between the two plants, supporting the specialisation of the only insect currently known as a pollinator of this *Ophrys*, suggests the existence of at least one other species of pollinator for this *Ophrys*, and the potential existence of different pollinators between the two subspecies of this *Ophrys* (Bertrand in prep.). In this way, the analysis of quantitative information derived from pictures taken in the field may help in focusing further investigations on unknown additional putative insect pollinators.

Our classifiers showed good performance, while we chose to keep it simple. Here, we (i) used the default setup and the simplest methods of each package to extract data, (ii) we had a relatively small sample size (52–57)individuals by subspecies), (iii) we kept all specimens in our analyses, even plants with slightly faded or damaged flowers, and (iv) we used random forests, which is the most simple machine learning algorithm available, requiring only balanced data. Other methods available in the literature may advantageously complement our approach, be it for extracting additional quantitative information—for instance, by extending the set of analyses of shapes (e.g., R package *momocs* [22]) or colour patterns (ImageJ plugin *pat-geom* [14]). Other machine learning classifiers are also available, with some already used in ecology and evolution, such as the support vector machines (SVM) algorithm or some belonging to deep learning (artificial neural network), showing better performance for species identification [8]. Interestingly, some of these methods are being tested on *Ophrys*, but currently only to test whether artificial intelligence may tease apart the *Ophrys* species flower and its known insect pollinator [56].

The results of our study also show that packages such as *colordistance* [25], *patternize* [23] and, to a lesser extent, *geomorph* [21] are fully relevant for plant models, while they were initially developed for (or using) animal models (fish, butterflies, salamanders, etc.). This being said, there are obvious limitations to information extracted from images as in our approach. For instance, here, we only considered colour information that can be perceived in the visible spectra (and encoded in the RGB colour space), which is different from what bees can actually perceive [57]. A more comprehensive characterisation of flower colour would require combined knowledge of chemistry (pigment composition and concentration) and physics (absorption, transmission, colour saturation, etc. [11]). Thus, colour from an image-based approach is a first step that needs to be further investigated. Moreover, soft plant structures may be inappropriate for geomorphometrics as they may impede the location of landmarks on homologous positions, thus providing noisy information [23]. This potentially explains the poor accuracy obtained for the classifier built exclusively on shape data.

## 5. Conclusions

Image-based approaches coupled with simple machine learning (random forest) algorithms are powerful to detect subtle phenotypic differentiation and correctly assign individuals to predefined groups. This analytical framework is therefore of great interest in fields such as integrative taxonomy or to identify potential traits for local adaptation and/or reproductive isolation in evolutionary ecology. It advantageously complements but does not replace other integrative taxonomy methods in such a way that it will not specifically tell the user what is the most likely number of groups (taxa) to best describe the data [58]. As long as the sampling design is balanced, random forest is a also flexible enough to incorporate any phenotypic data, as well as genotypic and or environmental data. This method must now be tested on other *Ophrys* species complexes to verify its effectiveness and consider the possibility of its generalisation, particularly on photo recognition tools such as Pl@ntNet ([7]; https://identify.plantnet.org/ accessed on 1 March 2022). The ultimate objective is to provide an operational identification tool for those involved in the conservation of these species, which are sometimes protected, as it is the case here for *Ophrys aveyronensis* , but this requires us to test the performance of random forest models on other populations. Otherwise, although the relevance of a trait for classification does not mean biological relevance, this approach is potentially fruitful to provide a list of potential traits. The eco-evolutionary relevance of these traits needs to be subsequently assessed through additional approaches, including outlier detection and association methods or $P_{ST}$-$F_{ST}$ comparisons, as for traits highlighted from more conventional approaches. In the specific case of *Ophrys*, and particularly of the *O. aveyronensis* species complex, this approach may thus certainly allow us to address new research avenues in biogeography and integrative taxonomy, especially in systems such as the *Ophrys aveyronensis* species complex, for which genotypic data are still scarce [59].

**Author Contributions:** Conceptualisation, J.A.M.B.; data acquisition, R.B., B.S., A.G. and J.A.M.B.; methodology, F.L., J.A.M.B., A.G. and M.B.; formal analysis, F.L., J.A.M.B. and A.G.; investigation, F.L. and J.A.M.B.; writing—original draft preparation, A.G. and J.A.M.B.; writing—review and editing, A.G., J.A.M.B., R.B., F.L., B.S. and M.B.; supervision, J.A.M.B.; project administration, J.A.M.B.; funding acquisition, J.A.M.B. All authors have read and agreed to the published version of the manuscript.

**Funding:** This research was funded by an ANR JCJC grant to J.B., grant number ANR-21-CE02-0022-01, and is set within the framework of the "Laboratoires d'Excellences (LABEX)" TULIP [ANR-10-LABX-41]. This work has been also supported by an inter-LabEX TULIP-CEMEB initiative (n°197310).

**Institutional Review Board Statement:** Not applicable.

**Informed Consent Statement:** Not applicable.

**Data Availability Statement:** All the datasets (phenotypic data, pictures, and metadata input files) used in the analyses presented in this article and the R code protocols used to run these analyses with these datasets are already freely accessible on the ZENODO [Open Aire EU official] repository: doi:10.5281/zenodo.6536168.

**Acknowledgments:** We thank Jean-Luc Roux and Daniel Vizcaïno for the information provided about sampling sites in Spain.

**Conflicts of Interest:** The authors declare no conflict of interest. The funders had no role in the design of the study; in the collection, analyses, or interpretation of data; in the writing of the manuscript, or in the decision to publish the results.

## Abbreviation

The following abbreviation is used in this manuscript:

RF     Random forest

## Appendix A

*Appendix A.1*

**Table A1.** Summary table of individuals and populations sampled for each subspecies of *O. aveyronensis.*

| Subspecies | Population | Nb of Individuals | Latitude | Longitude |
|---|---|---|---|---|
| *O. a.* subsp. *aveyronensis* | St-Affrique | 14 | 43.98 | 2.93 |
| *O. a.* subsp. *aveyronensis* | Lapanouse | 19 | 43.98 | 3.09 |
| *O. a.* subsp. *aveyronensis* | Guilhaumard | 19 | 43.84 | 3.19 |
| *O. a.* subsp. *vitorica* | Bercedo | 13 | 43.09 | -3.43 |
| *O. a.* subsp. *vitorica* | Larraona | 22 | 42.78 | -2.27 |
| *O. a.* subsp. *vitorica* | Valgañón | 22 | 42.31 | -3.08 |

**Table A2.** Summary table of identification criteria for *O. aveyronensis* subspecies from flora [39]. For comparisons, min and max of our measures are given in brackets.

| Criteria | *O. a.* subsp. *aveyronensis* | *O. a.* subsp. *vitorica* |
|---|---|---|
| Number of flowers | 5–8 sometimes 3–12 (2–10) | 2–5 (2–6) |
| Sepal length | 10–16 mm (8–13) | 8–10.5 mm (7–13.5) |
| Sepal width | 6–8 mm (3.5–7) | 3–5 mm (2.5–6) |
| Petal length | 7–10 mm (6–9) | 4–6.5 mm (4.5–9.5) |
| Petal width | 4–5.5 mm (2–6) | 2–4 mm (2–5) |
| Labellum length | 11–15 mm (8.5–15.5) | 10–13.5 mm (8–14) |
| Labellum width | 12–18 mm (7–13) | 11–15 mm (8–14) |
| Labellum shape and colour pattern | gibosity, hairiness, "marbled" and varied macula | "H-like" shape macula |
| Sepal colour | light to dark pink, rarely creamy white, exceptionally greenish | pink to pale purple |

*Appendix A.2*

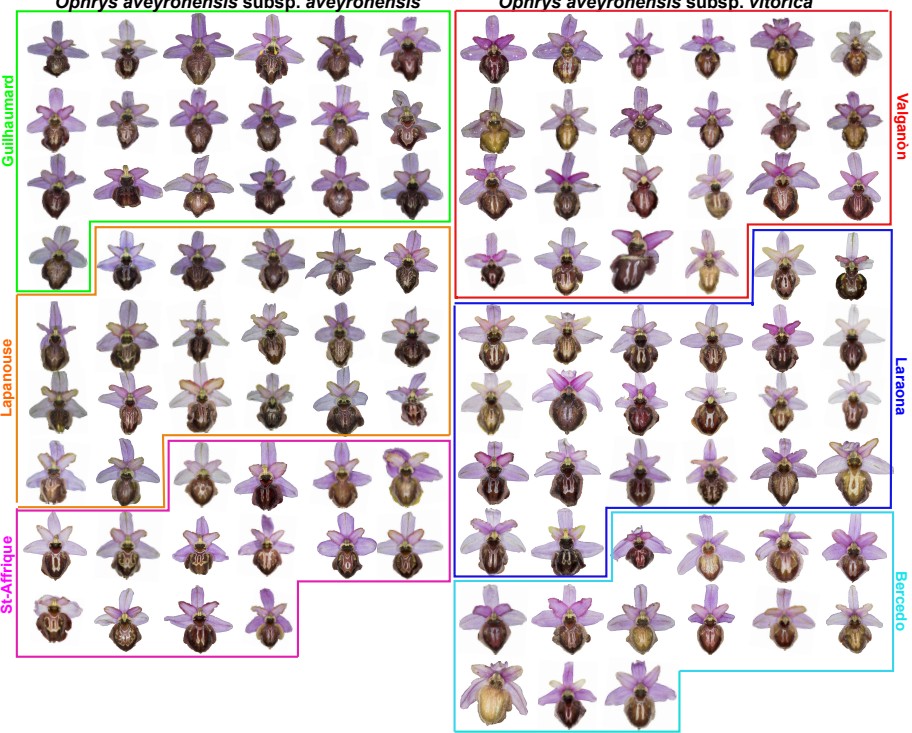

**Figure A1.** Pictures of *O. a.* subsp. *aveyronensis* and *O. aveyronensis* subsp. *vitorica* sorted by populations.

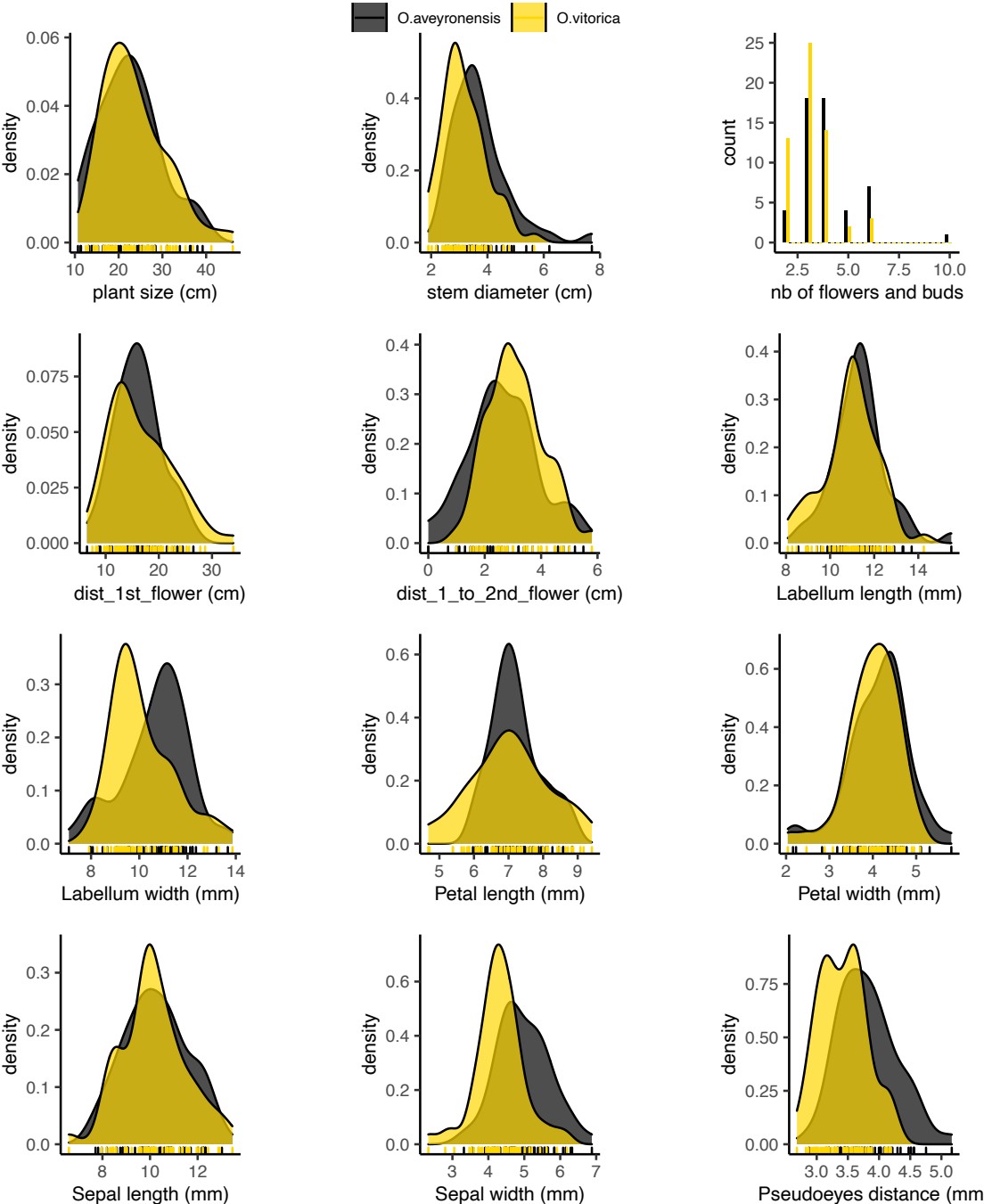

**Figure A2.** Distribution of morphological traits for *O. a.* subsp. *aveyronensis* and *O. aveyronensis* subsp. *vitorica*.

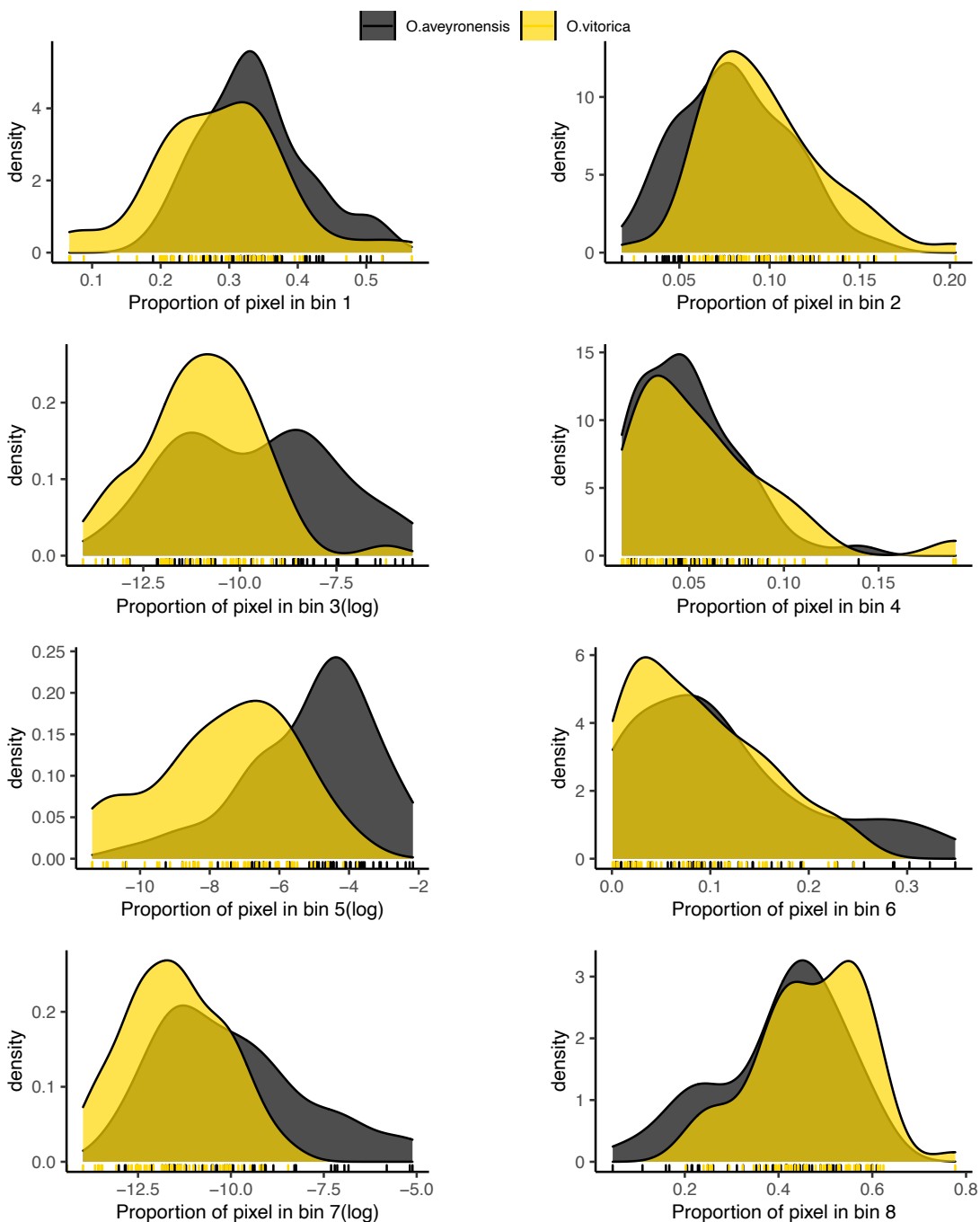

**Figure A3.** Distribution of colour traits for *O. a.* subsp. *aveyronensis* and *O. aveyronensis* subsp. *vitorica*.

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
