# Peer review of "Extracting Quantitative Information from Images Taken in the Wild: A Case Study of Two Vicariants of the Ophrys aveyronensis Species Complex"

_diversity, doi:10.3390/d14050400_

Round 1
Reviewer 1 Report
This is an interesting paper dealing with the application of image-based analyses and AI to distinguish between two putative orchid species. The idea is not very original as these approaches have represented the ground for the development of a long list of apps for plant/animal identification today available on mobile devises. Still the paper has some elements of novelty in trying to apply, in a rigorous scientific context, the approaches to a specific biological/taxonomic question to detect subtle morphological differences between two putative sister species/subspecies that are very hard to take apart. Overall, I think the work is methodologically correct and the authors have correctly acknowledged all limitations of their approach. I would only have preferred an independent validation of the efficiency of the method as the AI has been trained on the same population set but I can understand it was not easy to avoid this caveat with such endemic/localized species.
Below I offer a few suggestions/comments to improve the MS before acceptance:
Abstract, line 3: Why does it only apply to biogeography? This sounds as a general problem.
Abstract, line 15: “identification and highlight eco-evolutionary relevant candidate traits”. This is a bit too far away from the current objectives (see also my comments at the end of Conclusions).
M&M line 141: I must assume the species is not protected in Spain.
M&M line 184-187 The authors have previously pointed out that how much are important the flower traits is a quite debated topic, here it seems they have firm idea
M&M line 216. Fig.3: I think part of this picture ( i.e. some of the histograms showing the proportion of pixel assigned to each of eight bins) can be moved in Appendix and part of this latter (the most informative) can be moved to the main picture.
Results, line 325-326: this sentence is not clear to me.
Results, line 327-332: I wonder how much the results are independent from the fact the RF has been trained on the same dataset. How does it work on new, different populations?
Results, line 338-339: and why?
Results, line 345: there is a quite strong population effect. How do the authors explain this?
Discussion, line 387: alternatively, the same, common pollinator is geographically variable, and orchids are locally adapted.
Discussion, line 400: many Andrena (female individuals) are oligolectic and often orchids bloom before their host plants as they attract males.
Conclusions, line 449-onwards
I would deemphasize this conclusion as this is quite speculative compared to the presented results. We have no idea whether these traits reflect evolutionary or demographic population patterns nor we know how much they are fixed or variable in other not-tested populations. Of course, I agree that anything looks different between two species deserves further attention, but this is true for all species.
Reviewer 2 Report
The manuscript presents an interesting application of image-based approach for assessing the morphometric floral traits variation among vicariant species in the same complex of Ophrys (Orchidaceae) species. It is also well written and visually appealing. By integrating few comments below, the manuscript highly deserves to be accepted for publication.
Results and conclusions are of interest to biogeographical studies that are quite few for orchids.
Title: Please consider shortening the title by maintaining essential words only.
A point of concern that needs to be addressed by appropriately implementing the discussion regards the usage of colour as floral trait. Colour has been considered as a result of human perception. However, given that the species are pollinated by bees, flower colour should be treated as perceived by bee potential pollinators. This limitation of the study needs to be clarified and concisely argued in discussion. For a sexual pollination system, colour may play a secondary role in insect attraction, however, for the applicability as a broad scale approach, colour should be treated as perceived by bees (or other potential pollinators). As a reference to integrate to support this argument I recommend:
Lunau et al., 2021
Lunau, K., Scaccabarozzi, D., Willing, L., & Dixon, K. (2021). A bee’s eye view of remarkable floral colour patterns in the south-west Australian biodiversity hotspot revealed by false colour photography. Annals of botany, 128(7), 821-824.
Please include the following reference to support your statements:
L308: at the end of the sentence 'Scaccabarozzi et al., 2020'
Scaccabarozzi, D., Guzzetti, L., Phillips, R. D., Milne, L., Tommasi, N., Cozzolino, S., & Dixon, K. W. (2020). Ecological factors driving pollination success in an orchid that mimics a range of Fabaceae. Botanical Journal of the Linnean Society, 194(2), 253-269.
